# PLGA Nanoparticles Decorated with Anti-HER2 Affibody for Targeted Delivery and Photoinduced Cell Death

**DOI:** 10.3390/molecules26133955

**Published:** 2021-06-28

**Authors:** Victoria Olegovna Shipunova, Anna Samvelovna Sogomonyan, Ivan Vladimirovich Zelepukin, Maxim Petrovich Nikitin, Sergey Mikhailovich Deyev

**Affiliations:** 1Shemyakin–Ovchinnikov Institute of Bioorganic Chemistry, Russian Academy of Sciences, 16/10 Miklukho-Maklaya St., 117997 Moscow, Russia; annasogomonyan2012@mail.ru (A.S.S.); ivan.zelepukin@gmail.com (I.V.Z.); max.nikitin@gmail.com (M.P.N.); biomem@mail.ru (S.M.D.); 2Institute of Engineering Physics for Biomedicine (PhysBio), MEPhI (Moscow Engineering Physics Institute), 31 Kashirskoe Shosse, 115409 Moscow, Russia; 3Moscow Institute of Physics and Technology, 9 Institutskiy per., 141701 Dolgoprudny, Russia; 4Department of Nanobiomedicine, Sirius University of Science and Technology, 1 Olympic Ave, 354340 Sochi, Russia

**Keywords:** nanoparticles, PLGA, affibody, HER2, Rose Bengal, targeted delivery, reactive oxygen species

## Abstract

The effect of enhanced permeability and retention is often not sufficient for highly effective cancer therapy with nanoparticles, and the development of active targeted drug delivery systems based on nanoparticles is probably the main direction of modern cancer medicine. To meet the challenge, we developed polymer PLGA nanoparticles loaded with fluorescent photosensitive xanthene dye, Rose Bengal, and decorated with HER2-recognizing artificial scaffold protein, affibody Z_HER2:342_. The obtained 170 nm PLGA nanoparticles possess both fluorescent and photosensitive properties. Namely, under irradiation with the green light of 540 nm nanoparticles, they produced reactive oxygen species leading to cancer cell death. The chemical conjugation of PLGA with anti-HER2 affibody resulted in the selective binding of nanoparticles only to HER2-overexpressing cancer cells. HER2 is a receptor tyrosine kinase that belongs to the EGFR/ERbB family and is overexpressed in 30% of breast cancers, thus serving as a clinically relevant oncomarker. However, the standard targeting molecules such as full-size antibodies possess serious drawbacks, such as high immunogenicity and the need for mammalian cell production. We believe that the developed affibody-decorated targeted photosensitive PLGA nanoparticles will provide new solutions for ongoing problems in cancer diagnostics and treatment, as well in cancer theranostics.

## 1. Introduction

The development of new, and the improvement of existing methods of diagnostics and therapy of oncological diseases is impossible without the involvement of new non-standard tools in biology and biomedicine [1,2]. Nanoparticles of various natures have a wide range of properties that are not inherent in bulk samples or small molecules, thus making it possible to solve the most pressing problems both in therapy and in the diagnosis of diseases [3,4,5,6,7,8,9].

The use of nanoparticles as drug carriers significantly reduces the doses of substances injected into the bloodstream while maintaining or even increasing the therapeutic index. In particular, the clinic already has a number of nanoparticle-based drugs that deliver standard chemotherapeutic compounds to tumor cells, such as, for example, liposomal forms of doxorubicin (Caelyx) or Myocet [10]. Such drugs are accumulated in the tumor due to the damaged vasculature; this effect is called the enhanced permeability and retention (EPR) effect.

However, recent studies demonstrate that the EPR effect works in rodents but does not work in humans [11]. In this regard, there is a need to increase the effectiveness of nanomedications, which is achieved either by prolongation of the nanoparticles’ circulation time in the bloodstream by various methods [12,13,14,15] or by designing targeted drug delivery systems (DDSs) allowing for both the delivery and retention of drugs in the tumor.

For the targeted delivery of nanostructures of a different nature to cancer cells, different targeting molecules are used, mainly full-length immunoglobulins. However, the use of IgG for the delivery of nanopreparations has many serious drawbacks: high immunogenicity, the unoriented conjugation of molecules to the surface of the nanoparticle, and the need for the production of molecules in mammalian cell cultures due to the necessary post-translational modifications. Artificial scaffold polypeptides, which have already proven themselves as promising instruments for nanoparticle delivery, are an excellent alternative to full-length IgGs [16,17,18,19,20,21,22,23]. Such polypeptides, for example, DARPins and affibodies, are obtained by the method of ribosomal or phage display, and have dissociation constants in the picomolar range for certain specific targets. They are efficiently produced in bacterial systems, do not possess post-translational modifications, do not have cysteines in their structure, and both N- and C-terms are available for both chemically and genetically engineered manipulations. Moreover, they are of low immunogenicity and very small in size (8 kDa for affibody molecules).

Here, we developed polymer biodegradable nanoparticles loaded with xanthene dye and modified with a targeted affibody polypeptide. We used poly (d, l-lactide-co-glycolide) as a matrix for nanoparticle synthesis. These nanoparticles selectively interacted only with HER2-overexpressing cells and also produced reactive oxygen species (ROS) when exposed to green light irradiation, thereby causing cell death. Thus, we have obtained diagnostic (fluorescent) and inducible-therapeutic (ROS-producing) nanoparticles that selectively recognize the well-known human oncomarker HER2. The developed DDS is a step towards the development of new generation therapy and diagnostic nanoparticle-based agents.

## 2. Results

### 2.1. Synthesis and Modification of PLGA Nanoparticles

Polymer nanoparticles were synthesized, as schematically illustrated in Figure 1. Poly(d, l-lactide-co-glycolide) (PLGA, lactide: glycolide 50:50, acid and hydroxy-terminated, 25 kDa, Sigma, Germany) was used as a matrix for the nanoparticle synthesis with the “water-in-oil-in-water” method.

The first emulsion was made by the addition of Rose Bengal at 3 g/L to 300 µL of 40 g/L PLGA in chloroform and sonication with the 130 Watt Sonicator Vibra-Cell (Sonics) at +4 °C for 1 min (Figure 1a). The second emulsion was formed by dropping the first emulsion into 3 mL of 5% PVA (Mowiol^®^ 4-88, Sigma, Germany) in Milli-Q water with 1 g/L of chitosan oligosaccharide lactate (5 kDa, Sigma, Germany). The second emulsion was sonicated again for one minute. The resulting nanoparticles (Figure 1b) were washed by triple centrifugation with PBS and finally resuspended in 300 µL of PBS. The final concentration of the particles was determined by drying at 60 °C. Synthesized PLGA particles possess both -COOH (due to PLGA) and -NH_2_ (due to chitosan) chemical groups on the surface, which made their direct chemical conjugation to proteins, namely, affibody, possible. Finally, to specifically target the synthesized nanoparticles to HER2 positive cells, the nanoparticles were modified with HER2-recognizing affibody, namely, Z_HER2:342_, using the carbodiimide chemistry (Figure 1c).

The synthesized nanoparticles were characterized using scanning electron microscopy (Figure 2a). According to SEM image processing, these particles are monodisperse structures with a form close to the spherical. The real size of as-synthesized nanoparticles is equal to 170 ± 40 nm. The DLS measurements revealed that the hydrodynamic size of PLGA nanoparticles is similar, equal to 198 ± 40 nm, with a polydispersity index equal to PDI = 0.07 (Figure 2b). Visual observations revealed that particles were stable for one year (no further observations were made).

The effective loading of synthesized PLGA nanoparticles with fluorescent dye Rose Bengal was confirmed with the the optical and fluorescence spectroscopy methods. The absorbance spectrum of PLGA and Rose Bengal-loaded PLGA is presented in Figure 2c, thus confirming the Rose Bengal presence in the nanoparticle structure. To confirm that the Rose Bengal is indeed associated with PLGA, we performed a flow cytometry assay evaluating the fluorescence intensity of nanoparticle populations, considering PLGA particles as single events. The flow cytometry histograms of nanoparticles’ populations are presented in Figure 2d, thus proving the loading of Rose Bengal into the nanoparticle.

The excitation and emission spectra of nanoparticles presented in Figure 2e,f correspond to that of loaded dye with excitation and emission maximums of 560 nm, and 575 nm, respectively. Since Rose Bengal is a dye photoactivated with green light capable of producing reactive oxygen species (1O_2_, O_2_*^−^, H_2_O_2_ and HO*), to confirm the preservation of ROS generation ability within the architecture of polymer particles, particles were irradiated with the green light for 15 min, and general oxidative stress indicator CM-H_2_DCFDA was added to the tubes. The fluorescence intensities of samples were then measured at an excitation of 492 nm and an emission of 525 nm.

The obtained data confirmed the formation of ROS after the green light irradiation—the fluorescence intensity of samples after irradiation was 5.14 times higher than that of non-irradiated samples (4991 ± 101 a.u. for non-irradiated PLGA sample vs. 25673 ± 67 a.u. for irradiated PLGA sample).

### 2.2. HER2-Overexpressing Cell Targeting by PLGANanoparticles Modified with Anti-HER2 Affibody Z_HER2:342_


To realize the targeted delivery of the obtained nanoparticles to cells, we used small recognizing polypeptide, affibody Z_HER2:342_. Affibody Z_HER2:342_ is a non-IgG scaffold polypeptide possessing a high affinity constant of binding to HER2 equal to 22 pM [24,25].

For in vitro targeting experiments, two cell lines with different expression levels of HER2 receptor were selected: (i) human breast adenocarcinoma cell line SK-BR-3 with an overexpression of HER2 receptor; (ii) Chinese hamster ovary cell line CHO without the expression of HER2. First, cells were evaluated in terms of specific labeling with affibody Z_HER2:342_ recognizing HER2 receptor. For this aim, Z_HER2:342_ was conjugated with fluorescein isothiocyanate (FITC) to obtain Z_HER2:342_-FITC. Cells were labeled with Z_HER2:342_-FITC and analyzed with confocal microscopy. Data presented in Figure 3 confirm the specific labeling of cells with HER2 overexpression with this fluorescent protein conjugate.

To quantitatively estimate the specificity of this kind of binding, a flow cytometry assay was performed. As a positive control, the widely used anti-HER2 antibody conjugated to FITC was used, namely, Trastuzumab-FITC. The flow cytometry histograms corresponding to cells’ populations labeled with Z_HER2:342_-FITC are presented in Figure 4a, confirming the Z_HER2:342_-FITC specificity in terms of cell labeling. However, since HER2 is presented on normal cell lines to a much lesser extent, one more cell line possessing HER2 normal expression level, namely, human lung carcinoma A549, was added to the flow cytometry assay as well as for further experiments.

The next step of the study was directed to the creation of targeted PLGA particles. The modification of the PLGA nanoparticle surface with targeted affibody molecules was made with carbodiimide chemistry using EDC and sulfo-NHS as crosslinking agents. Affibody containing -COOH groups was activated with an excess of EDC/sulfo-NHS in acidic conditions and then added to nanoparticles containing -NH_2_ groups (due to the chitosan presence) in slightly alkaline conditions, thus resulting in the formation of a stable amide bond between protein and nanoparticle. The efficiency of covalent conjugation was confirmed with BCA protein assay, showing that the 5.4 μg of affibody molecules were bound to 1 mg of PLGA nanoparticle surface.

The resulting fluorescent targeted nanoparticles denoted as PLGA*Z_HER2:342_ were incubated with cells, cells were washed from non-bound particles and analyzed with flow cytometry in the fluorescent channel corresponding to the fluorescence of Rose Bengal.

The flow cytometry histograms corresponding to cells’ populations labeled with PLGA*Z_HER2:342_ are presented in Figure 4b. Indeed, the MFI of SK-BR-3 cells labeled with PLGA*Z_HER2:342_ is 2.13 times higher than the corresponding value for A549 cells and 4.22 times higher than for CHO cells (Figure 4c), thus quantitatively supporting the specificity of targeted nanoparticles binding. Thus, using the targeted scaffold protein, affibody Z_HER2:342_, and carbodiimide chemistry, we obtained HER2-specific nanoparticles capable of generating ROS under external irradiation with light and specifically targeting cancer cells.

### 2.3. Cellular Toxicity of Rose Bengal-Loaded PLGA Nanoparticles

The obtained PLGA particles could be effectively used as a nano-cargo of a photosensitizer in photodynamic therapy. To evaluate the light-induced toxicity of the as-synthesized particles, the MTT toxicity assay was carried out. Human breast cancer cells SK-BR-3 were incubated with different concentrations of PLGA, washed from non-bound particles, and irradiated with the green light for 5, 10, or 20 min. Next, after 48 h of cultivation, the cytotoxicity was measured using the tetrazolium dye MTT-3-(4,5-dimethylthiazol-2-yl)-2,5-diphenyltetrazolium bromide. Data presented in Figure 5a confirm the concentration-dependent cytotoxicity of PLGA under irradiation with green light with virtually total cell death after 20 min of irradiation for moderate concentrations of PLGA used for incubation with cells.

To evaluate the efficacy of the targeted nanoparticles for HER2-positive cancer cell elimination, for the control experiments, we used non-targeted PLGA loaded with Rose Bengal (Figure 5b), targeted PLGA without Rose Bengal (Figure 5c), and non-targeted PLGA without Rose Bengal (Figure 5d). Data presented in Figure 5 confirm that the most pronounced effect of cytotoxicity is observed for PLGA nanoparticles loaded with Rose Bengal and conjugated to the targeted molecule.

However, a small cytotoxic effect is observed for the Rose Bengal loaded PLGA*Z_HER2:342_ particles without irradiation, which is most probably caused by cells’ lipid peroxidation with a xanthene dye Rose Bengal. Alongside the particle-induced cytotoxicity, the green light-induced cytotoxicity is observed as well, which is most probably caused by ROS production, a change in intracellular calcium, and a significant drop in intracellular pH [26]. Thus, by combining the irradiation with light and the use of targeted therapeutic nanoparticles, it is possible to achieve the most intensive cytotoxic effect on the desired site of action.

## 3. Discussion

Modern methods of cancer treatment, such as chemotherapy or radiotherapy, have a wide range of side effects for humans, such as chronic cardiotoxicity, myelosuppression, hepatotoxicity, neurotoxicity, and a number of others [27,28,29].

The efforts of modern medicine are directed towards the development of targeted drug delivery systems (DDSs) in order to increase the effectiveness of treatment and minimize systemic side effects. Nanoparticles of various natures, in particular, polymeric ones, are the most promising platform for creating such DDSs. In particular, the clinic already has a number of medications based on polymer nanoparticles, approved for administration to humans [30].

Natural biocompatible polymers, e.g., proteins, chitosan, or biocompatible biosynthetic polymers, are the most successful matrix for the targeted delivery of diagnostic and therapeutic components for cancer cells targeting.

Here we used the PLGA polymer, a copolymer of lactic and glycolic acids, which is completely biocompatible and biodegradable and degrades to lactic and glycolic acids which are presented in the organism, thereby causing negligible side effects. It is worth noting that PLGA has already been approved by the FDA for therapeutic use in humans —there are more than fifteen PLGA-based formulations for the treatment of numerous disorders [31,32]. PLGA is one of the most popular polymers for the synthesis of nanoparticles with a therapeutic payload [33,34].

To incorporate therapeutic functions into the architecture of synthesized PLGA nanoparticles, we loaded PLGA with Rose Bengal. Rose Bengal (4,5,6,7-tetrachloro-2’,4’,5’,7’-tetraiodofluorescein) is a xanthene dye capable of producing reactive oxygen species under irradiation with green light and causing oxidative stress in cancer cells [35]. Rose Bengal has already proved itself as an efficient anti-cancer medication for dermal or subcutaneous melanoma metastases treatment in humans [36]. Moreover, it is commonly used for staining damaged conjunctival and corneal cells within the composition of eye drops [37,38], thus confirming the validity of the use of Rose Bengal in living systems.

Next, we equipped the developed nanoparticles with a targeting modality capable of specifically targeting HER2-overexpressing cancer cells. HER2 is a clinically relevant oncomarker that is overexpressed in 30% of human breast carcinomas and its expression often correlates with the high metastatic potential of tumor and poor prognosis. Thus, this receptor serves as a clinically relevant target on the surface of cancer cells.

To target the developed photosensitive PLGA particles to HER2-overexpressing cancer cells, we decorate them with a small polypeptide scaffold-affibody Z_HER2:342_. Affibody molecules are a novel class of recognizing proteins of non-immunoglobulin origin, possessing a wide range of advantages for applications in the clinic. These molecules are originally based on the Z domain (the immunoglobulin G binding domain) of protein A and are made via the randomization of 13 amino acids located in two alpha-helices involved in the target binding in the original protein [39,40]. In addition to their small size (6–8 kDa, respectively), these molecules exhibit high affinity to molecular targets, low immunogenicity, excellent solubility, and stability, with both N- and C-terms available for chemical conjugation and genetic engineering manipulations. It should be also noted that these molecules already entered the clinical trials, thus proving the effectiveness of such a class of antibody mimetics: e.g., anti-HER2 (111)In-ABY-025 appears to be safe for use in humans and effective for discriminating HER2 status in metastatic breast cancer, regardless of ongoing HER2-targeted antibody treatment [41].

## 4. Materials and Methods

### 4.1. Electron Microscopy

Scanning electron microscopy images of PLGA nanoparticles were obtained with a MAIA3 electron microscope (Tescan, Brno—Kohoutovice, Czech Republic) at an accelerating voltage of 10 kV. The samples were deposited onto a silicon wafer and then air-dried at ambient conditions. SEM images were evaluated using ImageJ software to obtain a mean particle size.

### 4.2. Fluorescence Spectroscopy

PLGA nanoparticle excitation and emission spectra were acquired using an Infinite M100 Pro (Tecan, Grödig, Austria) microplate reader. The nanoparticle suspension in 100 µL of PBS at a concentration of 10 µg/mL was placed into a 96-well flat-bottomed plate. The excitation spectrum was recorded within the range of 350–600 nm (emission wavelength was 670 nm) at the bottom mode. The emission spectrum was recorded within the range of 550–850 nm (excitation wavelength was 530 nm) at the bottom fluorescence mode.

### 4.3. Protein Quantification

The quantity of affibody conjugated to PLGA nanoparticles was estimated using the BCA Protein Assay kit (Pierce), according to the manufacturer’s recommendations. Affibody-conjugated and pristine PLGA nanoparticles were incubated with BCA (bicinchoninic acid) solution for 30 min at 37 °C. The absorbance was measured at 562 nm. The quantity of protein per 1 mg of PLGA nanoparticles was calculated using the calibration curve obtained using samples with serial dilutions of free affibody incubated with BCA solution for 30 min at 37 °C.

### 4.4. Dynamic Light Scattering Measurements

The hydrodynamic sizes of nanoparticles were determined using a Zetasizer Nano ZS (Malvern Instruments Ltd., Enigma Business Park, Grovewood Road, Malvern, UK) analyzer in PBS buffer (137 mM NaCl, 2.7 mM KCl, 4.77 mM Na_2_HPO_4_, 1.7 mM KH_2_PO_4_, pH 7.4) at 25 °C. All measurements were performed in triplicate.

### 4.5. Cell Culture

Cell lines of human breast adenocarcinoma SK-BR-3 (HTB-30; ATCC), human lung carcinoma A549 (CCL-185; ATCC) and Chinese hamster ovary CHO (Russian Cell Culture Collection) were maintained in DMEM medium (HyClone, Logan, Utah, USA) supplemented with 10% fetal bovine serum (HyClone, Logan, UT, USA) and 2 mM L-glutamine (PanEko, Moscow, Russia). Cells were incubated under a humidified atmosphere with 5% CO_2_ at 37 °C.

### 4.6. Protein Conjugation to FITC

Proteins were conjugated to FITC as described by us previously [17,18,19].

### 4.7. Confocal Laser Scanning Microscopy

For Z_HER2:342_-FITC visualization, cells were labeled with 5 µg/mL Z_HER2:342_-FITC and Hoechst 33342 (1 µg/mL) on ice for 30 min in PBS with 1% BSA, washed from unbound molecules and imaged at the following conditions: Hoechst 33342: excitation laser—405 nm, emission filter—445/45 nm; FITC: excitation laser—488 nm, emission filter—525/45 nm.

### 4.8. Cytotoxicity Assay

The cytotoxicity of nanoparticles was evaluated using the MTT test. SK-BR-3 cells were incubated with nanoparticles for 30 min, irradiated with 540 nm laser, and seeded onto a 96-well plate at 5 × 10^3^ cells per well in 200 μL of DMEM medium supplemented with 10% FBS (heat-inactivated fetal bovine serum). The cells were cultured for 48 h at 37 °C with 5% CO_2_. Then, the medium was removed and 100 μL of MTT solution (tetrazolium dye, 3-(4,5-dimethyl-2-thiazolyl)-2,5-diphenyl-2*H*-tetrazolium bromide) at 0.5 g/L in DMEM medium was added to the wells. Cells were incubated for 1 h at 37 °C with 5% CO_2_. The MTT solution was removed, and 100 μL of DMSO (dimethyl sulfoxide) was added to the wells; the plate was gently shaken until the formazan crystals were completely dissolved. The optical density of the wells was measured with a microplate reader Infinite M100 Pro (Tecan, Grödig, Austria) at a wavelength of λ = 570 nm with reference λ = 630 nm. All samples were performed in triplicate.

### 4.9. PLGA Conjugation with Z_HER2:342_

The covalent modification of PLGA nanoparticles with affibody was carried out using EDC (Sigma, Darmstadt, Germany) and sulfo-NHS (Sigma, Darmstadt, Germany) as crosslinking agents via the formation of amide bonds between the carboxyl groups of affibody and amino groups of PLGA. An amount of 100 µg of affibody (purified as described by us previously [17]) in 100 μL of 0.1 M 2-(N-morpholino) ethanesulfonic acid buffer, pH 5.0 was activated with 50 µg EDC and 25 µg sulfo-NHS for 45 min at 15 °C. Next, the activated affibody Z_HER2:342_ was quickly added to 1 mg of PLGA nanoparticles in 300 μL of borate buffer (0.4 M H_3_BO_3_, 70 mM Na_2_B_4_O_7_, pH 8.0) and sonicated for several seconds. The reaction was carried overnight at room temperature, followed by particle washing from non-bound proteins by triple centrifugation at 4000× *g* for 10 min.

### 4.10. Flow Cytometry

To determine Z_HER2:342_-FITC and PLGA*Z_HER2:342_ specificity, the harvested cells were washed with PBS, resuspended in 300 µL of PBS with 1% BSA at a concentration of 10^6^ cells per mL, labeled with Z_HER2:342_-FITC at a final concentration of 2 µg/mL or PLGA nanoparticles at 0.1 g/L, and washed and analyzed using the Novocyte 3000 VYB flow cytometer (ACEA Biosciences, San Diego, CA, USA) in the BL1 channel (excitation laser 488 nm, emission filter 530/30 nm) for Z_HER2:342_-FITC and the YL2 channel (excitation laser 561 nm, emission filter 615/20 nm) for PLGA nanoparticles.

### 4.11. Statistical Analysis

All assays were performed in triplicate. The statistical significance of the experimental results was determined with a two-tailed Student’s *t*-test (*p* **  <  0.001).

## 5. Conclusions

We developed targeted PLGA nanoparticles loaded with xanthene dye, Rose Bengal, directed toward HER2-overexpressing cancer cells. We showed that under irradiation with external green light, these nanoparticles generate ROS, leading to cancer cell death. All the components of the developed drug delivery system are already approved for clinical applications in humans, thus confirming the possible rapid translation of the developed system into clinical practice for diagnostic and therapeutic applications in cancer medicine. We believe that the developed system will provide new, promising solutions for ongoing problems in cancer treatment.

## Figures and Tables

**Figure 1 molecules-26-03955-f001:**
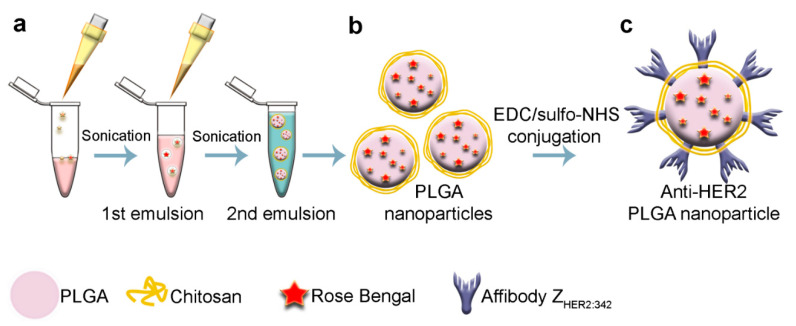
Schematic illustration of PLGA nanoparticle synthesis and chemical modification. (**a**) PLGA nanoparticles were synthesized by the “water-in-oil-in-water” emulsion method. Rose Bengal in water was introduced into the solution of PLGA in chloroform and emulsified by sonication. The first emulsion was introduced into the solution of PVA with chitosan oligosaccharide lactate and once again emulsified. The chloroform was then evaporated and as-obtained PLGA nanoparticles (**b**) were washed from PVA with centrifugation. (**c**) PLGA nanoparticles were conjugated to affibody Z_HER2:342_ via carbodiimide chemistry using EDC (1-ethyl-3-(3-dimethyl aminopropyl) carbodiimide hydrochloride) and sulfo-NHS (*N*-hydroxysulfosuccinimide) as crosslinking agents.

**Figure 2 molecules-26-03955-f002:**
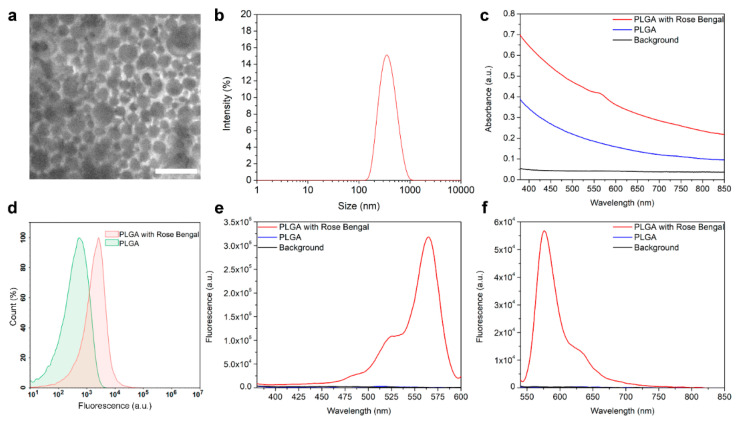
Characterization of PLGA nanoparticles. (**a**) Scanning electron microscopy image of as-synthesized PLGA nanoparticles. (**b**) Hydrodynamic size distribution of PLGA nanoparticles obtained from DLS measurements. (**c**) The absorbance of as-synthesized pristine PLGA nanoparticles and Rose Bengal-loaded PLGA nanoparticles. (**d**) Flow cytometry assay on evaluation of fluorescence of PLGA nanoparticles loaded with Rose Bengal in the fluorescence channel corresponding to the Rose Bengal fluorescence. Excitation (**e**) and emission (**f**) spectra of the PLGA and Rose Bengal-loaded PLGA nanoparticles.

**Figure 3 molecules-26-03955-f003:**
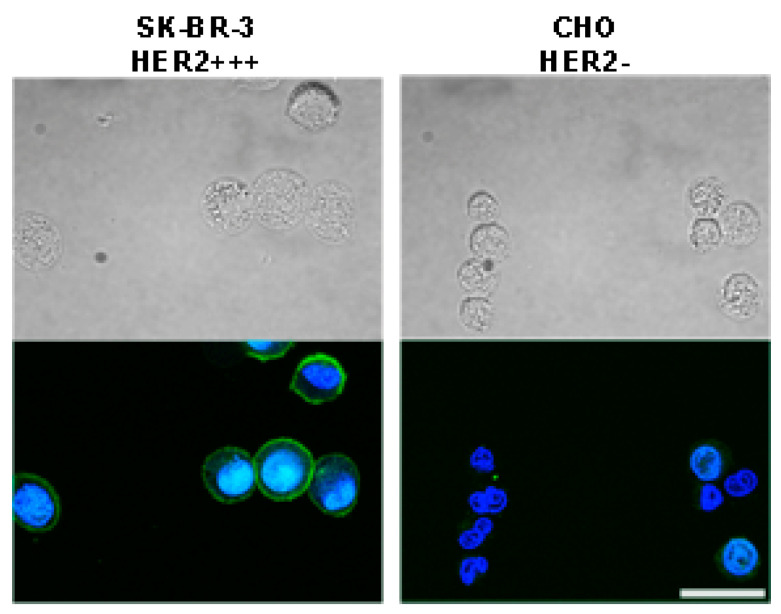
Fluorescent immunostaining of cells possessing different HER2 expression levels with affibody Z_HER2:342_. Affibody Z_HER2:342_ was conjugated with FITC and cells were labeled with Z_HER2:342_-FITC. Top panels show bright-field images of SK-BR-3 and CHO cells, and bottom panels present overlaid confocal images of cells incubated with Z_HER2:342_-FITC and Hoechst 33342 (Hoechst 33342: excitation laser 405 nm, emission filter 445/45 nm; Z_HER2:342_-FITC: excitation laser 466 nm, emission filter 525/45 nm). Scale bar, 25 µm.

**Figure 4 molecules-26-03955-f004:**
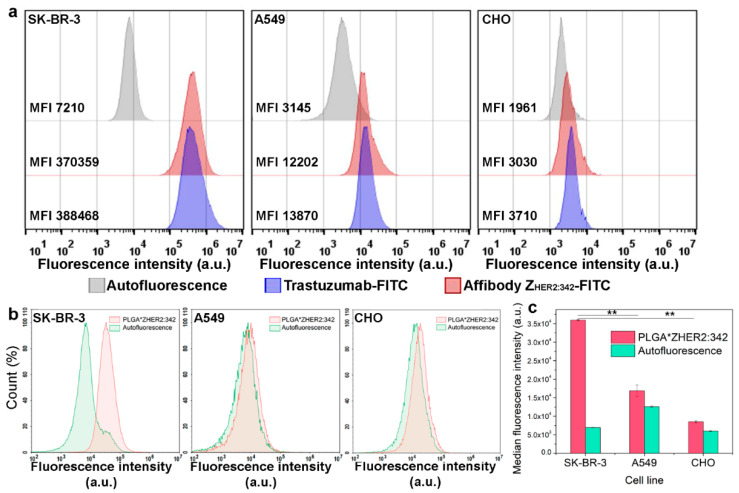
Flow cytometry assays on assessing the specificity of affibody Z_HER2:342_ and nanoparticles modified with affibody, PLGA* Z_HER2:342_. (**a**) Cells were labeled with Trastuzumab-FITC and Z_HER2:342_-FITC and analyzed in BL1 channel (excitation laser 488 nm, emission filter 530/30 nm). Autofluorescence is shown by grey, cells labeled with Trastuzumab-FITC are shown by blue, with Z_HER2:342_-FITC are shown by red. (**b**) Flow cytometry assay: evaluation of PLGA*Z_HER2:342_ nanoparticles bound to cells. Flow cytometry histograms were acquired in the YL2 channel (excitation laser 561 nm, emission filter 615/20 nm). Autofluorescence is shown by green, cells labeled with PLGA*Z_HER2:342_ nanoparticles are shown by red. (**c**) Median fluorescence intensities of cells’ populations labeled with PLGA*Z_HER2:342_ in the corresponding fluorescent channel of the flow cytometer. ** *p* < 0.001.

**Figure 5 molecules-26-03955-f005:**
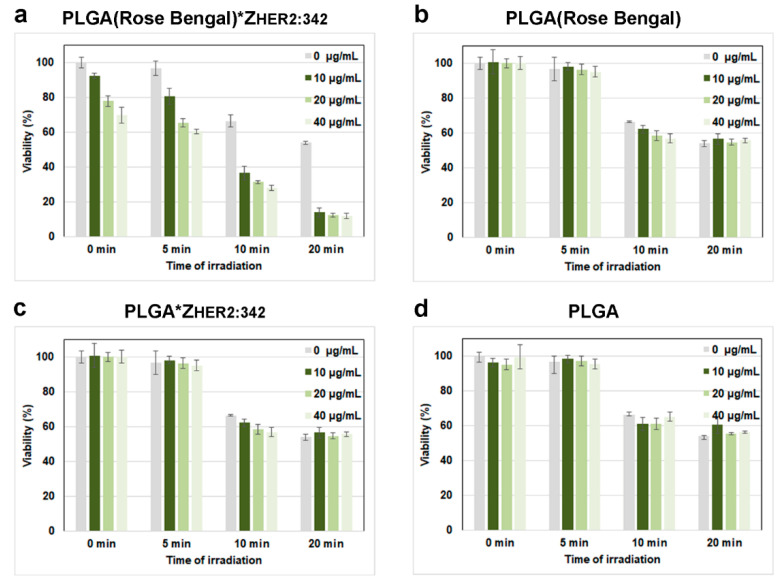
Cellular toxicity of as-synthesized PLGA nanoparticles. The results of the MTT toxicity test for cells incubated with different concentrations of PLGA, washed from non-bound particles and irradiated with the green light for 0, 5, 10, and 20 min for Rose Bengal-loaded PLGA*Z_HER2:342_ (**a**) Rose Bengal-loaded PLGA. (**b**) PLGA*Z_HER2:342_. (**c**) PLGA nanoparticles. (**d**) The data presented are the percentage of survived cells after 48 h of incubation in comparison with control non-treated cells.

## Data Availability

The data presented in this study are available on request from the corresponding author.

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
