# Peer review of "PLGA Nanoparticles Decorated with Anti-HER2 Affibody for Targeted Delivery and Photoinduced Cell Death"

_molecules, 2021, doi:10.3390/molecules26133955_

Round 1
Reviewer 1 Report
The manuscript “PLGA Nanoparticles Decorated with Anti-HER2 Affibody for the Targeted Delivery and Photoinduced Cell Death” proposes a nanoparticle system aimed at targeted drug delivery and photoinduced therapy of breast cancer. The methods of nanoparticle preparation, Rose Bengal encapsulation and affibody modification are presented, as well as some studies in vitro with cell line(s).
It is claimed that PLGA-based nanoparticles were loaded with Rose Bengal and modified with an affibody, and that “Chemical conjugation of PLGA with anti-HER2 affibody resulted in selective binding of nanoparticles only to HER2-overexpressing cancer cells.” However, as detailed below, the experimental data is not sufficient or clear enough to support those conclusions.
Major issues
1-Drug encapsulation and Peptide modification of the nanoparticles:
The results in section 2.1/figure 2.C suggest that Rose Bengal is associated to the PLGA nanoparticles, but the spectra of simple/blank PGLA nanoparticles is missing for comparison. The fluorescence microscopy or flow cytometry techniques could give further evidence that the drug is in fact encapsulated into the particles or surface-associated.
Importantly, the manuscript shows no results evidencing that the affibody peptide is in fact covalently linked to the nanoparticles.
2-The cytotoxicity of blank PLGA supports:
Cytotoxicity of PLGA nanoparticles without Rose Bengal and affibody is missing. Blank PLGA nanoparticles prepared in the same way (including for example EDC, but without the drug and affibody), can show cytotoxicity and increase with irradiation. Indeed, irradiation per si seems cytotoxic (approx. 50% reduction in Figure 3). In the absence of data for blank supports it is not possible to assess the value of the modified nanoparticles.
The nanoparticles up to 1 g/L caused more than 50% cytotoxicity, without irradiation (Figure 3). A reason or commentary for this observation?
3) The targeting efficiency of the synthesized particles
The (selective) targeting of HER2-expressing cells by the nanoparticles is highlighted in Abstract and Conclusions. Indeed, targeting abilities can be important for therapeutic efficacy. However, the data in Figure 4 indicates a selectivity of the affibody but there are no results for the proposed nanoparticles. And the data in Figure 5 indicates that the nanoparticles deliver Rose Bengal to both cell lines, significantly different of the highly selective affibody interaction results. Once again there is no data for Blank nanoparticles.
Minor points
-Abstract: The sentence “HER2…belongs to the EGFR family…” should be reformulated for correctness.
-Methods: Please avoid using “RT”.
-Discussion: The sentence “…these molecules already entered the clinical trials [39] thus proving the effectiveness…” should be reformulated if their effectiveness is still under evaluation/trial.
Author Response
The authors’ team is very grateful for the valuable comments and thanks to the reviewers for helping to improve the quality of the manuscript.
Please find below the point-by-point reply for these comments.
Comment 1:
Drug encapsulation and Peptide modification of the nanoparticles: The results in section 2.1/figure 2.C suggest that Rose Bengal is associated to the PLGA nanoparticles, but the spectra of simple/blank PGLA nanoparticles is missing for comparison. The fluorescence microscopy or flow cytometry techniques could give further evidence that the drug is in fact encapsulated into the particles or surface-associated. Importantly, the manuscript shows no results evidencing that the affibody peptide is in fact covalently linked to the nanoparticles.
Response 1:
The absorbance spectra, fluorescence excitation and emission spectra of Rose Bengal-loaded PLGA (including blank particles PLGA and background), as well as flow cytometry histograms, were obtained and presented in Figure 2 thus confirming the efficient dye (Rose Bengal) loading. The efficiency of covalent conjugation was confirmed with BCA protein assay showing that the 5.4 μg of affibody molecules were bound to 1 mg of PLGA nanoparticle surface.
Comment 2:
The cytotoxicity of blank PLGA supports:
Cytotoxicity of PLGA nanoparticles without Rose Bengal and affibody is missing. Blank PLGA nanoparticles prepared in the same way (including for example EDC, but without the drug and affibody), can show cytotoxicity and increase with irradiation. Indeed, irradiation per si seems cytotoxic (approx. 50% reduction in Figure 3). In the absence of data for blank supports it is not possible to assess the value of the modified nanoparticles. The nanoparticles up to 1 g/L caused more than 50% cytotoxicity, without irradiation (Figure 3). A reason or commentary for this observation?
Response 2:
All the mentioned in the reviewer’s comment control experiments were performed, including the MTT assay on assessing the cytotoxicity of blank PLGA, PLGA with affibody without Rose Bengal, and PLGA without affibody and with Rose Bengal were performed and incorporated into the main text (Fig. 5). The results are supported with the explanation of the mechanism of the cytotoxicity of targeted nanoparticles without the irradiation as well as the intrinsic green light-induced cytotoxicity.
Comment 3:
The targeting efficiency of the synthesized particles. The (selective) targeting of HER2-expressing cells by the nanoparticles is highlighted in Abstract and Conclusions. Indeed, targeting abilities can be important for therapeutic efficacy. However, the data in Figure 4 indicates a selectivity of the affibody but there are no results for the proposed nanoparticles. And the data in Figure 5 indicates that the nanoparticles deliver Rose Bengal to both cell lines, significantly different of the highly selective affibody interaction results. Once again there is no data for Blank nanoparticles.
Response 3:
The targeting efficiency of as-obtained PLGA nanoparticles conjugated with anti-HER2 affibody was evaluated in detail and presented in Fig. 4b and Fig. 4c; the results are supported with their explanation in the manuscript main text. Moreover, one cell line possessing a normal HER2 expression level was added to the assay thus proving the specificity of HER2-directed PLGA nanoparticles.
Comment 4:
Abstract: The sentence “HER2…belongs to the EGFR family…” should be reformulated for correctness.
Response 4:
Corrected.
Comment 5:
Methods: Please avoid using “RT”.
Response 5:
Corrected.
Comment 6:
Discussion: The sentence “…these molecules already entered the clinical trials [39] thus proving the effectiveness…” should be reformulated if their effectiveness is still under evaluation/trial.
Response 6:
Corrected to “It should be also noted that these molecules already entered the clinical trials thus proving the effectiveness of such a class of antibody mimetics: e.g., anti-HER2 (111)In-ABY-025 appears to be safe for use in humans and effective for discriminating HER2 status in metastatic breast cancer, regardless of ongoing HER2-targeted antibody treatment [40]”.
Reviewer 2 Report
This interesting paper "PLGA Nanoparticles Decorated with Anti-HER2 Affibody for the Targeted Delivery and Photoinduced Cell Death" by Shipunova and co-authors present affibody-decorated targeted photosensitive PLGA nanoparticles as a tool for diagnostics and treatment of cancers and for cancer theranostics. The idea of manuscript is innovative, obtained results are promising and prospective. Therefore, I recommend this paper for publication in the “Molecules”, section: Medicinal Chemistry, special Issue "Bioactive Peptides and Proteins". Additionally, I have several suggestions to improve the manuscript, as noted below.
- The authors do not provide the description of the method that has been used for the statistical analysis of the tests performed. Statistical analysis/significance should be carried out as well.
- Authors claim that studied nanoparticles are monodisperse. But they do not report PDI values.
- In the experiments of targeting effects authors compare the human cells with the hamster cells. Since these cells are completely different, such comparison is not appropriate.
- Authors wrote that fluorescent probe CM-H2DCFDA was used to measure the level of ROS. However they give the absorbance values on the graph. The description of the axes in the chart should be corrected or explained.
- In the “Materials and methods” section, authors state that the MTT test was performed on both cell lines while results show SKBR cells only. The text of “Materials and methods section” should be rewritten appropriately or mentioned results must be included in the manuscript.
- The quality of fig. 2 should be improved.
- In the figure 3b the error bar (0.5 g/L) is not visible.
- The small insignificant errors should be corrected, for example line 123 the CM-H2DCFDA instead CM-H2DCFDA, line 156, 324, 340, 378 in vitro should be given in italics, typos such as "labeled" instead "labelled" etc.
Author Response
The authors’ team is very grateful for the valuable comments and thanks to the reviewers for helping to improve the quality of the manuscript.
Please find below the point-by-point reply for these comments.
Comment 1:
The authors do not provide the description of the method that has been used for the statistical analysis of the tests performed. Statistical analysis/significance should be carried out as well.
Response 1:
The statistical analysis description was added to the main text.
Comment 2:
Authors claim that studied nanoparticles are monodisperse. But they do not report PDI values.
Response 2:
PDI value was calculated: “the DLS measurements revealed that the hydrodynamic size of PLGA nanoparticles is similar and equal to 198 ± 40 nm with polydispersity index equal to PDI = 0.07 (Fig. 2b)”.
Comment 3:
In the experiments of targeting effects authors compare the human cells with the hamster cells. Since these cells are completely different, such comparison is not appropriate.
Response 3:
To make the comparison more eligible, we added one more cell line to the targeting experiments, namely, the human lung carcinoma A549 cell line possessing normal HER2 expression level.
Comment 4:
Authors wrote that fluorescent probe CM-H2DCFDA was used to measure the level of ROS. However they give the absorbance values on the graph. The description of the axes in the chart should be corrected or explained.
Response 4:
The description of the experiment was corrected.
Comment 5:
In the “Materials and methods” section, authors state that the MTT test was performed on both cell lines while results show SKBR cells only. The text of “Materials and methods section” should be rewritten appropriately or mentioned results must be included in the manuscript.
Response 5:
The description of the experiment was corrected.
Comment 6:
The quality of fig. 2 should be improved.
Response 6:
SEM imaging was performed once again and high-quality image was obtained and presented in Fig.2a.
Comment 7:
In the figure 3b the error bar (0.5 g/L) is not visible.
Response 7:
The quality of the figure was improved with the substantial amount of additional experiments.
Comment 8:
The small insignificant errors should be corrected, for example line 123 the CM-H2DCFDA instead CM-H2DCFDA, line 156, 324, 340, 378 in vitro should be given in italics, typos such as "labeled" instead "labelled" etc.
Response 8:
Corrected.
Round 2
Reviewer 1 Report
The revised manuscript “PLGA Nanoparticles Decorated with Anti-HER2 Affibody for the Targeted Delivery and Photoinduced Cell Death” incorporated new data, namely on Blank PLGA nanoparticles. These new data gives support to the Conclusions regarding the nanoparticles loading with Rose Bengal and modification with peptide, their targeting and cytotoxicity properties.
The method for the BCA protein assay applied to the nanoparticles is missing, and the caption for Figure 4.b should be revised as the plots don’t show data for PLGA nanoparticles.
Author Response
Comment1: “The method for the BCA protein assay applied to the nanoparticles is missing”
Reply1: The BCA protein assay method was added to the Methods section as follows:
«4.3. Protein quantification
The quantity of affibody conjugated to PLGA nanoparticles was estimated using the BCA Protein Assay kit (Pierce) according to tot the manufacturer’s recommendations. Affibody-conjugated and pristine PLGA nanoparticles were incubated with BCA (bicinchoninic acid) solution for 30 min at 37 °C. The absorbance was measured at 562 nm. The quantity of protein per 1 mg of PLGA nanoparticles was calculated using the calibration curve obtained using samples with serial dilutions of free affibody incubated with BCA solution for 30 min at 37 °C».
Comment2: “the caption for Figure 4.b should be revised as the plots don’t show data for PLGA nanoparticles”
Reply2: Corrected.
Reviewer 2 Report
I am completely satisfied by authors' answers to my questions. The paper deserves to be published as it is.
Author Response
N/A (all comments were addressed previously).